# Study of the Normal Crested Porcupine (*Hystrix cristata*) Nasal Cavity and Paranasal Sinuses by Cross-Sectional Anatomy and Computed Tomography

**DOI:** 10.3390/vetsci11120611

**Published:** 2024-11-29

**Authors:** Daniel Morales Bordon, Francisco Suárez-Cabrera, Gregorio Ramírez, Pablo Paz-Oliva, Alejandro Morales-Espino, Alberto Arencibia, Mario Encinoso, Myriam R. Ventura, José Raduan Jaber

**Affiliations:** 1Departamento de Patología Animal, Producción Animal, Bromatología y Tecnología de Los Alimentos, Facultad de Veterinaria, Universidad de Las Palmas de Gran Canaria, Trasmontaña, Arucas, 35413 Las Palmas, Spain; daniel.morales@ulpgc.es (D.M.B.); myriam.rodriguezventura@ulpgc.es (M.R.V.); 2Departamento de Morfología, Facultad de Veterinaria, Universidad de Las Palmas de Gran Canaria, Trasmontaña, Arucas, 35413 Las Palmas, Spain; ecovetcanarias@yahoo.es (F.S.-C.); pablo.paz101@alu.ulpgc.es (P.P.-O.); alejandro.morales108@alu.ulpgc.es (A.M.-E.); alberto.arencibia@ulpgc.es (A.A.); 3Departamento de Anatomía y Anatomía Patológica Comparadas, Facultad de Veterinaria, Universidad de Murcia, 30100 Murcia, Spain; grzar@um.es; 4Hospital Veterinario, Facultad de Veterinaria, Universidad de Las Palmas de Gran Canaria, Trasmontaña, Arucas, 35413 Las Palmas, Spain; 5VETFUN, Educational Innovation Group, University of Las Palmas de Gran Canaria, Trasmontaña, Arucas, 35413 Las Palmas, Spain

**Keywords:** head, computed tomography, anatomy, nasal cavity, paranasal sinuses, crested porcupine

## Abstract

The nasal cavity involves essential functions in rodents, including olfaction, respiration, behavior, and reproduction. The crested porcupine (*Hystrix cristata*) is a species of rodent native to Italy, North Africa, and sub-Saharan Africa, and it is listed as “least concern” by the International Union for Conservation of Nature (IUCN). However, in porcupines, there is little information available about the anatomy of this species. Therefore, we conducted a study using cross-sectional anatomy and computed tomography (CT) to highlight the specific anatomical findings of the nasal cavities, paranasal sinuses, and associated structures of the crested porcupine. The results obtained here could be used by biologists, researchers, and practitioners to learn about the organization of the structures that constitute this region.

## 1. Introduction

The crested porcupine (*Hystrix cristata*) is a well-known member of the family Hystricidae. Since 2016, it has been on the IUCN red list as the least concerning [1]. It occurs mainly in Europe and Africa, where these animals are still hunted illegally for meat and killed because they are considered a pest species due to the damage produced to crops [2]. Consequently, porcupines require adequate conservation policies in regional and local contexts. It is native to Italy and Sicily, inhabiting dry shrubland, maquis, abandoned farmland, steppe, forest, and rocky areas. There is limited information on its geographic variability in Africa [1,2,3], where these animals are found in various habitats such as forest formations, woodland savannahs, mountains, croplands, and sandhill deserts. They seek refuge in caves, aardvark holes, and burrows that they dig themselves [1,2].

*Hystrix cristata*, also known as histrik, are large rodents measuring 60–90 cm in length and 25 cm in height. Their bodies are covered in long black-and-white-striped spikes up to 30 cm long, dotted with shorter and thicker ones covering the entire back and tail. The quills that run along the head, nape, and back can be raised into a crest, which gives them the name of a crested porcupine. Moreover, they come off and are projected as a defense mechanism [4]. This porcupine has a gestational period of approximately 112 days, where one or two offspring of both sexes are born [5]. These babies typically reach adult weight at one to two years of age, and just before then, they reach sexual maturity [4]. Porcupines are strictly nocturnal, walking long distances for food [4,6]. They are herbivorous and consume plant material [6,7], including roots, tubers, bark, rhizomes, bulbs, cultivated crops and fallen fruits. This plant material represents a significant portion of their diet [8].

Regarding its anatomical head remarks, it shows a large and round head with small ears. Its skull presents a large infraorbital foramen and a remarkable nasal cavity development, and the angular process is inflected on the lower jaw [4]. Moreover, histrik shows notable prominences in the skull used for masticatory muscle attachment [7]. In addition, it shows a well-developed incisor, two premolars, and two molars in each quadrant [4].

The anatomical intricacy between the different mammal species and the expanding attentiveness to wildlife species have represented a challenge to practitioners in evaluating imaging techniques studies [9]. Thanks to technological advancements, obtaining anatomical and diagnostic information has become easier and much faster. Therefore, diagnostic techniques have become remarkable tools for medical practice, combining conventional methods, such as radiology and ultrasounds [9], with advanced imaging procedures, like CT or magnetic resonance imaging (MRI) [9,10,11,12,13,14,15]. These modern techniques offer better imaging resolution, fast image acquisition, and the clear visualization of structures, thus providing detailed anatomical and functional information with an improved capability to differentiate between bone and soft tissue formations.

Thus far, various studies on the anatomy and pathology of pet rabbits and rodents are already available using diagnostic techniques [15,16,17,18,19,20,21]. However, to our knowledge, no research has been conducted on the crested porcupine’s nasal cavities and paranasal sinuses using anatomical cross-sections and CT images. The rodent’s nasal cavity is an anatomical structure with multiple functions related to the respiratory and olfactory systems. It is responsible for heating, humidifying, filtering the inspired air, and correctly directing it in the respiratory airways [22]. Considering the large size of this animal and its biology, eating plant material that they smell from the ground, we wanted to perform a deep description of this cavity and check any specific anatomical differences compared to mammals, including other rodents. This information could help to understand its respiratory and olfactory functions better.

## 2. Materials and Methods

### 2.1. Animals

Five adult crested porcupine carcasses (two males and three females) were obtained from the “Rancho Texas Lanzarote Park” in the Canary Islands, Spain. The animals had been euthanized from causes unrelated to this region. Neither porcupine had a history of nasal disease, and pathologic conditions were not detected during physical and radiologic examination.

### 2.2. CT Technique

The imaging study was conducted at the Veterinary Hospital of Las Palmas de Gran Canaria University using a 16-slice helical CT scanner (Toshiba Astelion, Canon Medical System, Tokyo, Japan). The five porcupines were placed in ventral recumbency on the CT scan table. The CT scan produced sequential transverse images with a thickness of 1 mm using a standard clinical protocol (100 kVp, 80 mA, 512 × 512 acquisition matrix, 1809 × 858 field of view, a spiral pitch factor of 0.94, and a gantry rotation of 1.5 s). To distinguish the CT appearance of the nasal cavity, two CT windows with different widths and levels were used: a bone window setting (WW = 1500; WL = 300), and a pulmonary window setting (WW = 1400; WL = −500). In addition, dorsal and sagittal multiplanar reconstructed (MPR) images were also obtained to enhance the identification of normal porcupine anatomical structures. All the CT images were imported to an image viewer (OsiriX MD, Apple, Cupertino, CA, USA) for data manipulation and analysis.

### 2.3. Anatomic Evaluation

We obtained transverse anatomical cross-sections of three animals after the scanning to help us identify the structures in the CT images. The animals were placed in a plastic holder, frozen (−80 °C), and then sliced into 1 cm sections using an electric band saw. These sections, thicker than the CT images, were cleaned with water, numbered, photographed, and compared with the corresponding CT view. We selected the sections that best matched the CT images to identify the relevant structures of the porcupine’s sinuses and nasal passages. In addition, we used anatomical texts and reputable references to aid in our identification process [14,15,16,20,22,23,24,25,26,27,28,29,30,31,32,33].

## 3. Results

No anatomic differences were identified in the heads of the five porcupines in this study. Images displaying relevant anatomical structures of the porcupine nasal cavity and paranasal passages are presented (Figure 1, Figure 2, Figure 3, Figure 4, Figure 5, Figure 6, Figure 7, Figure 8, Figure 9, Figure 10, Figure 11 and Figure 12). Figure 1 is a sagittal CT image, with lines and numbers (I-XI) indicating approximately the level of the following anatomical and transverse CT images. Figure 2, Figure 3, Figure 4, Figure 5, Figure 6, Figure 7, Figure 8, Figure 9, Figure 10, Figure 11 and Figure 12 consist of three transverse images: (A) an anatomical cross-section, (B) a bone CT window image, and (C) a lung CT window image. The images progress from the nose (Figure 2) to the ethmoidal labyrinth (Figure 12).

### 3.1. Anatomical Sections

These images displayed relevant formations of the nasal cavity and paranasal sinuses. Consequently, its rostral aperture was depicted by the nares, which had an oval shape. The outer more-or-less flaring wall of them is the radix nasi (root of the nose) that expands rostroventrally as the dorsum nasi (dorsum of the nose) until it is ended in the apex nasi (tip of the nose). Moreover, these sections showed the philtrum with adequate detail. Hence, it was presented as a vertical indentation with a Y-shape between the base of the nose and the border of the upper lip. Inside the nostrils, we identify a small-dilated area forming the nasal vestibule. More ventrally, we identified the incisive bone bearing the upper incisive teeth (Figure 2A, Figure 3A, Figure 4A, Figure 5A, Figure 6A and Figure 7A) and caudally, we identified the nasal septum, protected by a mucous membrane, and joined dorsally to the nasal bone (Figure 4A, Figure 5A, Figure 6A, Figure 7A, Figure 8A, Figure 9A and Figure 10A).

This septum shows a ventral part supported by the vomer (Figure 5A, Figure 6A, Figure 7A, Figure 8A, Figure 9A, Figure 10A and Figure 11A). Besides, we could identify the vomeronasal organ located ventrally to the vomer and it showed as a large-caliber duct Figure 4A, Figure 5A, Figure 6A, Figure 7A, Figure 8A and Figure 9A). The nasal septum divides the nasal cavity into two symmetrical passages and is prolonged dorso- and ventrolaterally, forming the dorsal and ventrolateral nasal cartilages (Figure 3A and Figure 4A). These transverse images were pivotal for identifying the straight, alar, basal (with its medial and lateral parts), and parallel folds (Figure 2A, Figure 3A, Figure 4A and Figure 5A). Caudally to the passages, we identified the nasal conchae represented by the dorsal, ventral, and medial nasal conchae (Figure 5A, Figure 6A, Figure 7A, Figure 8A, Figure 9A and Figure 10A). Between the walls of the nasal cavity and the nasal conchae, we distinguished different meatuses, including the dorsal, the middle, the ventral, and the common meatuses (Figure 6A, Figure 7A, Figure 8A, Figure 9A, Figure 10A and Figure 11A).

Additionally, these cross sections allowed the distinction of the conchomaxillary opening, connecting the middle nasal concha with the caudal compartment of the maxillary sinus (illustrated in Figure 10A). More caudal sections identified the fundus of the nasal cavity represented by the ethmoidal bone complex pinpointed between the neurocranium and the splanchnocranium, displaying a clear distinction between its perpendicular and tectorial plates. Besides, these sections allowed the identification of scroll-like plates of bone termed ethmoturbinates, which were differentiated between ectoturbinates and endoturbinates (Figure 11A). Other structures associated with the nasal cavity were the rostral part of the maxillary sinus, which communicated to the nasal cavity by the nasomaxillary opening through the ventral nasal meatus (Figure 7A and Figure 9A), the rostral and caudal compartments of the frontal sinus, and the palatine and sphenoidal parts of the sphenopalatine sinuses (Figure 9A, Figure 10A, Figure 11A and Figure 12A).

In addition, these images facilitate the depiction of structures belonging to the oral cavity, including the tongue, the lower incisors, the upper and lower lips, and the oral vestibule. Adjoining formations, including the nasopharynx, the hard palate, and the palatine plexus, were also depicted (Figure 4A, Figure 5A, Figure 6A, Figure 7A, Figure 8A, Figure 9A, Figure 10A, Figure 11A and Figure 12A). Furthermore, other notable bones, including the maxilla and the mandible bearing the premolar and molar teeth were displayed in these anatomical sections (Figure 8A, Figure 9A, Figure 10A and Figure 11A).

In the last anatomical images, we identify some brain structures, including the olfactory bulb and the olfactory recess (Figure 12A). Moreover, there is a clear distinction between the eye and its associated structures. Therefore, we distinguished the vitreous chamber, the sclera and the optic nerve, and the dorsal and ventral rectus muscles (Figure 11A). Moreover, we observed pivotal muscles associated with the masticatory activity, including the lateral and medial pterygoid muscles, the temporalis, the superficial and deeper portions of the masseter, and the buccinator muscle (Figure 9A, Figure 10A, Figure 11A and Figure 12A).

### 3.2. Computed Tomography (CT)

No relevant anatomic variations were observed in the five porcupines scanned. The anatomic formations displayed with the pulmonary and bone tissue matched adequately with structures in the complementary anatomical sections. Therefore, these CT windows showed a lateral cleft corresponding to the nostrils (Figure 2B,C). Moreover, the pulmonary and bone CT windows distinguished the philtrum as an I-shape in the ventral part of the face. This section was also essential in identifying the alar groove (Figure 2B,C and Figure 3B,C). More caudally, we observed different cartilages of the nose, including the dorsal and ventral lateral cartilages, the lateral accessory nasal cartilage, and various nasal folds, such as the alar, basal, straight, and parallel folds that showed intermediate attenuation signals (Figure 2B,C, Figure 3B,C, Figure 4B,C and Figure 5B,C). Enclosed by the nasal cartilages, we distinguished the nasal vestibule that showed hypoattenuation.

In contrast, this section displayed the incisive bone and the lower incisors with a high degree of attenuation. More caudal sections allowed the visualization of other bones that constitute the nasal cavity, including the nasal, the palatine, the maxillary, the vomer, and the ethmoid bones showing hyperattenuating signal (Figure 3B,C, Figure 4B,C, Figure 5B,C, Figure 6B,C, Figure 7B,C, Figure 8B,C, Figure 9B,C, Figure 10B,C, Figure 11B,C and Figure 12B,C). In addition, most parts of the nasal cavities and paranasal sinuses were displayed with the corresponding CT sections, which were easily identified due to the intraluminal gas content effect. Therefore, we distinguished the oral cavity, the nasopharyngeal duct, as well as the dorsal, ventral, and medial nasal conchal recesses, the maxillary, the frontal, and the sphenopalatine sinuses (Figure 8B,C, Figure 9B,C, Figure 10B,C, Figure 11B,C and Figure 12B,C).

Moreover, there were areas of soft tissue attenuation that were closely related to the mandible and compatible with the mylohyoideus, the buccinators, the temporal, lateral and medial pterygoideus, and the superficial and deeper portions of the masseter muscles (Figure 9B,C, Figure 10B,C, Figure 11B,C and Figure 12B,C). On these CT images, it was possible to visualize some adjacent formations that remained on the skull, including the olfactory bulb and diverse components of crested porcupine eyeballs, such as the extraocular muscles, the sclera, and the lens, appearing all of them as moderate to high attenuating structures (Figure 11B,C and Figure 12B,C).

## 4. Discussion

At present, the application of traditional imaging techniques, including radiology and ultrasound, is still quite helpful and very employed in exotic referral centers. Nonetheless, advanced imaging diagnostic techniques have proved essential in the anatomical knowledge and the diagnosis of different disorders in veterinary medicine [10,11,12,13,14,15,16,17,18]. These procedures give exceptional images of anatomic structures, an excellent definition of the extent and character of various pathologies, fast imaging acquisition and the avoidance of superimposition, which have revolutionized teaching purposes, research, and veterinary diagnosis [10,11]. Despite these advantages, the use of these procedures in exotic animal medicine is limited due to their costs, availability, and logistical challenges in obtaining images of captive and free-ranging animals. While several studies have been conducted using CT and MRI to assess the nasal cavities and paranasal sinuses in traditional and various exotic species, including domestic mammals, sea turtles, rabbits, koalas, guinea pigs, six-banded armadillos, and felids [17,20,27,28,29,30,31,32], there is a lack of an anatomical description of normal crested porcupine images. In this context, this study aims to provide anatomical details of the crested porcupine’s nasal cavity through anatomic sections and their correlation with transverse bone and pulmonary CT images. Other reports have used dorsal and sagittal images to obtain a complete visualization of the nasal cavities [29,30,34]. However, adequate information on the entire crested porcupine nasal cavity was also possible with those images obtained via the transverse sections. Relevant investigations have already demonstrated how the transverse plane has been helpful in veterinary medicine in identifying the nasal cavity and paranasal sinuses in detail, which is essential to evaluating nasal pathologic changes [29,31,35,36].

Concerning the nose, relevant structures were confirmed using these techniques. Since the nasal cavities and paranasal sinuses do not contain soft-tissue structures other than the mucosa, using different CT windows was quite helpful in evaluating the normal structures that compose this cavity. CT is considered the best imaging technique for an initial evaluation of the nasal cavity due to the excellent visualization of the bony limits and their extensions [34]. Other advantages of CT over MRI are lower cost and less time to conduct an examination. In contrast to CT, MRI allows differentiation between the nasal mucosa and other soft tissues or fluids, which can be valuable in interpreting pathological conditions [17,20,34]. In our work, we were working with animals without nasal pathologies. Thus, this fact and the combination of different CT windows and anatomical cross-sections provided excellent detail of this cavity. Therefore, we could distinguish various parts constituting the nose, which were better visualized in the anatomical cross-sections.

In contrast, the CT images provided excellent detail of the alar folds that showed relevant development in this species. Similar results were reported in exotic species such as the lion [34]. Nonetheless, other related species, including the six-banded armadillo, guinea pigs, koalas, rabbits, and big felids, such as the leopard and cheetah, did not report this extension [17,20,28,30,34]. As in lions [34], the caudal portion of this fold allowed the visualization of the incisive bone bearing the upper incisors.

Since we aimed to obtain detailed anatomical images with notable resolution, we performed our CT examinations with bone and pulmonary windows. The use of both windows in combination with the anatomical cross-sections provided pivotal details on the nasal conchae and the adjoining paranasal sinuses. Thus, we could distinguish a very little dorsal nasal concha, and a large ventral nasal concha divided into a dorsal and a ventral part, showing an oval shape and extending to the level of the first maxillary tooth. Important development of the nasal concha has been reported in other species, including domestic mammals, rabbits, guinea pigs, and six-banded armadillos [17,20,24,30,31,37]. Moreover, crested porcupines showed a notable development of the middle nasal concha compared to other wildlife and domestic species [34,37,38], depicted in the anatomical cross sections.

As in other species, such as the guinea pigs and the rabbits [20,37], we highlight the remarkable development of different paranasal sinuses. Therefore, we underlined the maxillary sinus, the largest paranasal sinus in the crested porcupine. This sinus was divided into rostral and caudal compartments. The rostral compartment was elongated in shape and extended from the upper incisive teeth roots to the level of the first premolar teeth. In contrast, the caudal compartment of the maxillary sinus was larger, and triangular, extending caudoventrally around the nasopharynx until the second molar teeth. According to other studies performed on this region [37,39], we used the cheek teeth to help in the sinus extension. Concerning the frontal sinus, it also reached proportionally large dimensions, extending under the parietal bone. Cheetahs and cats can also get large dimensions [38], although not as observed in the crested porcupines. Interestingly, other rodents do not show such extension of the frontal sinus [13,16,22,24]. All these sinuses serve several important functions. The mucosal lining helps warm and humidify the inhaled air from the nasal cavity. Additionally, these sinuses lighten the overall mass of the skull. This weight reduction allows the muscles responsible for moving the head and neck to operate more efficiently [16,22,34]. Here, we also highlighted that porcupines showed remarkable similarities with the equine paranasal sinuses [31,39]. Therefore, these rodents also presented a clear sphenopalatine sinus, which differentiated a palatine compartment with a triangular shape and a rounded sphenoidal compartment.

The information obtained about the crested porcupine nasal cavity and paranasal sinuses could be relevant for detecting pathological alterations reported in rodents and rabbits, including masses, foreign bodies, and acquired dental diseases and their associated consequences, such as deformities and osteomyelitis, and the extension of the infection process to different parts of the nasal and paranasal cavities [13,15,19,35,36].

The present investigation shows some limitations related to the evolution of sexual size dimorphism in favor of females in this animal. This has been related to the distribution and quality of food resources [40]. Despite this, we did not observe the essential difference between normal crested porcupine females and male nasal cavity samples. Nonetheless, further studies with more animals should be done to evaluate this possible difference and assess the notable development of the nasal cavity and paranasal sinuses observed in these animals.

## 5. Conclusions

This investigation provides a comprehensive anatomical description of the crested porcupine’s nasal cavity and paranasal sinuses using anatomical cross-sections and transverse CT images. The images obtained in this research adequately identified the anatomical landmarks of the rostral part of the porcupine head. Besides, these modern techniques provide essential insights to facilitate teaching applied anatomy as these imaging techniques provide excellent identification of specific anatomic structures without overlapping, enhancing the visualization and the organization of the nasal cavity in the crested porcupine. In addition, the details displayed here could be used as a preliminary source for the practitioner to evaluate CT images of porcupines with pathological conditions of the nasal cavity and paranasal sinuses.

## Figures and Tables

**Figure 1 vetsci-11-00611-f001:**
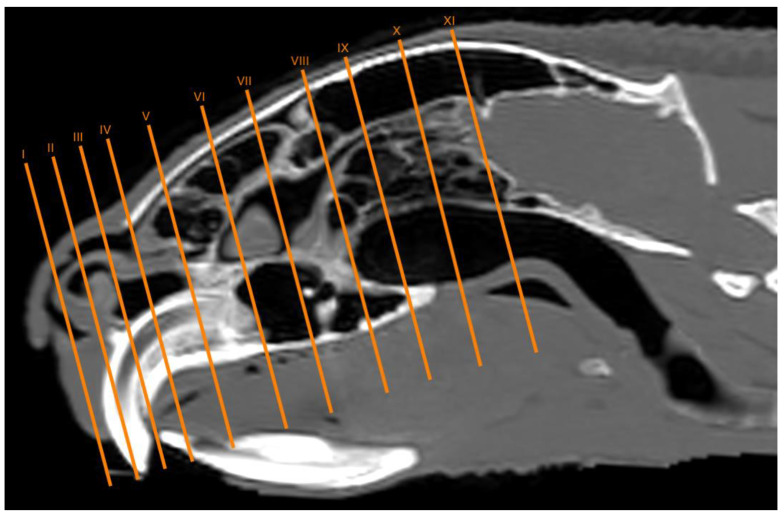
Parasagittal MPR CT image of the crested porcupine head depicting the approximate anatomical levels of the sections of the porcupine head. Sections I–XI correspond to Figure 2, Figure 3, Figure 4, Figure 5, Figure 6, Figure 7, Figure 8, Figure 9, Figure 10, Figure 11 and Figure 12.

**Figure 2 vetsci-11-00611-f002:**
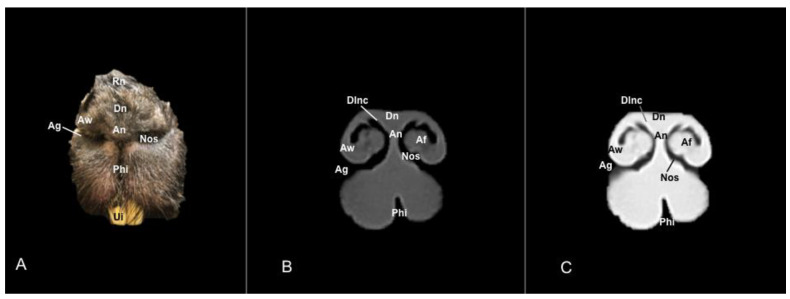
Anatomical section (**A**), bone window (**B**), and pulmonary window (**C**) CT transverse images of a crested porcupine’s nasal cavity at the level of the nose, corresponding to line I in Figure 1. Af: alar fold. Ag: alar groove. An: tip of the nose. Aw: alar wing. Dlnc: dorsal lateral nasal cartilage. Dn: dorsum of nose. Nos: nostrils. Phi: philtrum. Rn: root of nose. Ui: upper incisive tooth.

**Figure 3 vetsci-11-00611-f003:**
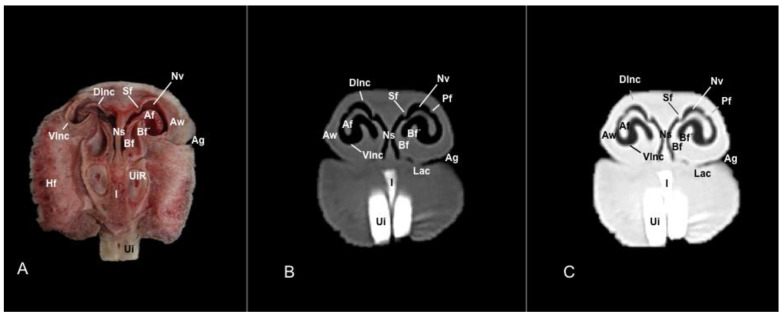
Anatomical section (**A**), bone window (**B**), and pulmonary window (**C**) CT transverse images of a crested porcupine’s nasal cavity at the level of the nasal vestibule, corresponding to line II in Figure 1. Af: alar fold. Ag: alar groove. Aw: alar wing. Bf: basal fold (medial part). Bf’: basal fold (lateral part). Dlnc: dorsal lateral nasal cartilage. Hf: hair follicle. I: incisive bone. Lac: lateral accessory nasal cartilage. Ns: nasal septum. Nv: nasal vestibule. Pf: parallel fold. Sf: straight fold. Ui: upper incisive tooth. UiR: upper incisive tooth root. Vlnc: ventral lateral nasal cartilage.

**Figure 4 vetsci-11-00611-f004:**
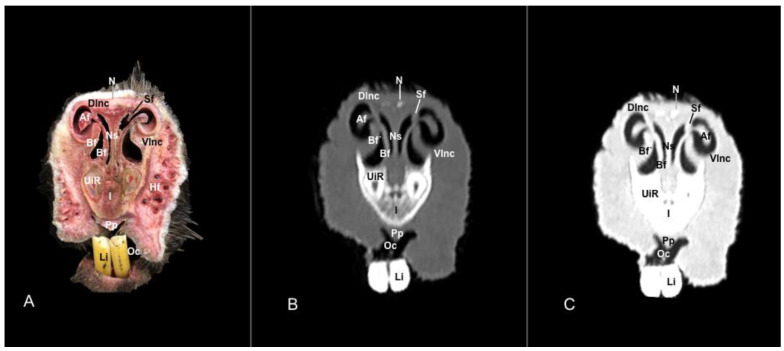
Anatomical section (**A**), bone window (**B**), and pulmonary window (**C**) CT transverse images of a crested porcupine’s nasal cavity at the level of the basal folds, corresponding to line III in Figure 1. Af: alar fold. Bf: basal fold (medial part). Bf′: basal fold (lateral part). Dlnc: dorsal lateral nasal cartilage. Hf: hair follicle. I: incisive bone. Li: lower incisive tooth. N: nasal bone. Ns: nasal septum. Nv: nasal vestibule. Oc: oral cavity. Pf: parallel fold. Pp: palatine plexus. UiR: upper incisive tooth root. Vlnc: ventral lateral nasal cartilage.

**Figure 5 vetsci-11-00611-f005:**
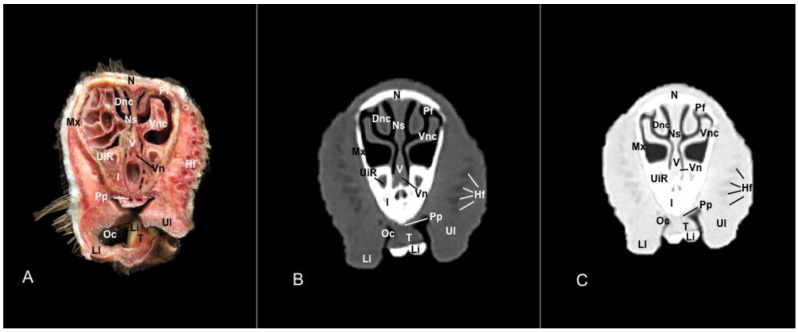
Anatomical section (**A**), bone window (**B**), and pulmonary window (**C**) CT transverse images of a crested porcupine’s nasal cavity at the level of the dorsal nasal concha, corresponding to line IV in Figure 1. Dnc: dorsal nasal concha. Hf: hair follicle. I: incisive bone. Li: lower incisive tooth. Ll: lower lip. Mx: maxilla. N: nasal bone. Ns: nasal septum. Oc: oral cavity. Pf: parallel fold. Pp: palatine plexus. T: tongue. Ul: upper lip. UiR: upper incisive tooth root. V: vomer. Vn: vomeronasal organ. Vnc: ventral nasal concha.

**Figure 6 vetsci-11-00611-f006:**
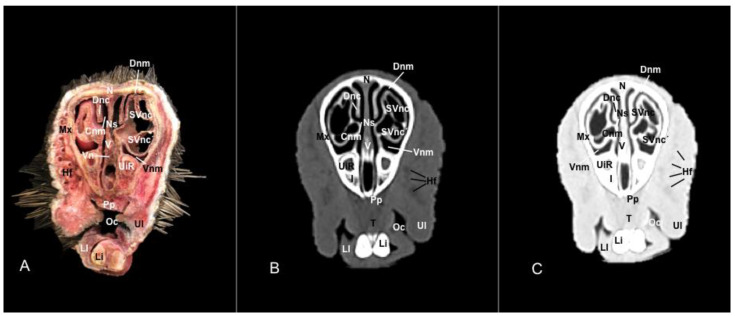
Anatomical section (**A**), bone window (**B**), and pulmonary window (**C**) CT transverse images of a crested porcupine’s nasal cavity at the level of the dorsal and ventral nasal conchae, corresponding to line V in Figure 1. Cnm: common nasal meatus. Dnc: dorsal nasal concha. Dnm: dorsal nasal meatus. Hf: hair follicle. I: incisive bone. Li: lower incisive tooth. Ll: lower lip. Mx: maxilla. N: nasal bone. Ns: nasal septum. Oc: oral cavity. Pp: palatine plexus. UiR: upper incisive tooth root. SVnc: sinus of ventral nasal concha (dorsal part). SVnc’: sinus of ventral nasal concha (ventral part). T: tongue. Ul: upper lip. V: vomer. Vn: vomeronasal organ. Vnm: ventral nasal meatus.

**Figure 7 vetsci-11-00611-f007:**
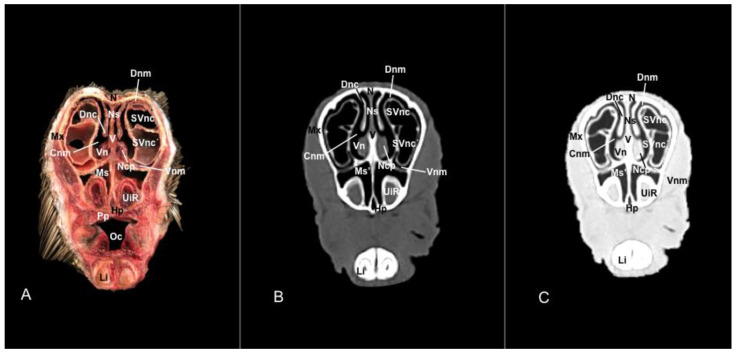
Anatomical section (**A**), bone window (**B**), and pulmonary window (**C**) CT transverse images of a crested porcupine’s nasal cavity at the level of the dorsal and ventral nasal conchae, and rostral maxillary sinus, corresponding to line VI in Figure 1. Cnm: common nasal meatus. Dnc: dorsal nasal concha. Dnm: dorsal nasal meatus. Li: lower incisive tooth. Mx: maxilla. Ms’: rostral maxillary sinus. N: nasal bone. Ncp: nasal cavernous plexuses. Ns: nasal septum. Oc: oral cavity. Pf: parallel fold. Pp: palatine plexus. SVnc: sinus of ventral nasal concha (dorsal part). SVnc’: sinus of ventral nasal concha (ventral part). UiR: upper incisive tooth root. V: vomer. Vn: vomeronasal organ. Vnm: ventral nasal meatus.

**Figure 8 vetsci-11-00611-f008:**
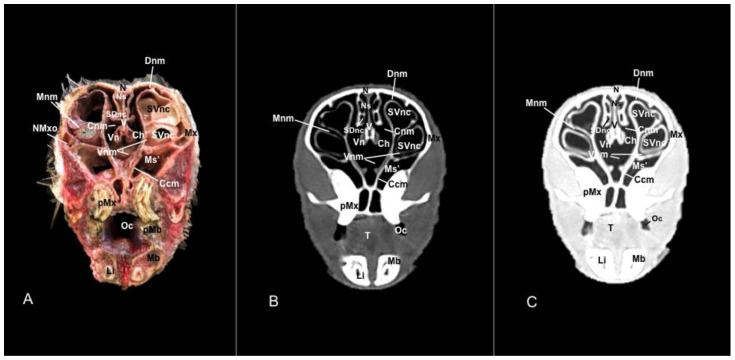
Anatomical section (**A**), bone window (**B**), and pulmonary window (**C**) CT transverse images of a crested porcupine’s nasal cavity at the level of the choanae, corresponding to line VII in Figure 1. Ch: choanae. Ccm: conchal crest of maxilla. Cnm: common nasal meatus. Dnm: dorsal nasal meatus. Li: lower incisive teeth. Mx: maxilla. Mb: mandible. Mnm: middle nasal meatus. Ms’: rostral maxillary sinus. N: nasal bone. Ns: nasal septum. NMxo: nasomaxillary opening. Oc: oral cavity. PMx: first premolar tooth (maxillar). PMb: first premolar tooth (mandibular). SDnc: sinus of dorsal nasal concha. SVnc: sinus of ventral nasal concha (dorsal part). SVnc’: sinus of ventral nasal concha (ventral part). V: vomer. Vn: vomeronasal organ. Vnm: ventral nasal meatus.

**Figure 9 vetsci-11-00611-f009:**
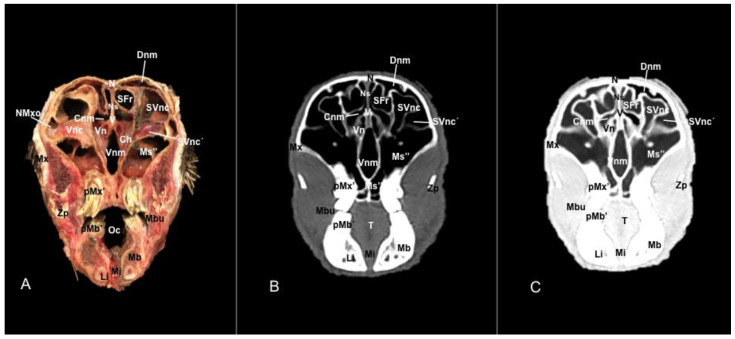
Anatomical section (**A**), bone window (**B**), and pulmonary window (**C**) CT transverse images of a crested porcupine’s nasal cavity at the level of the nasopharynx, corresponding to line VIII in Figure 1. Ch: choanae. Cnm: common nasal meatus. Dnm: dorsal nasal meatus. Li: lower incisive teeth. Mi: mylohyoid muscle. Mx: maxilla. Mb: mandible. Mbu: buccinator muscle. Ms’’: caudal maxillary sinus. N: nasal bone. Ns: nasal septum. NMxo: nasomaxillary opening. Oc: oral cavity. PMx’: second premolar tooth (maxillar). PMb’: second premolar tooth (mandibular). SFr: sinus of frontal bone (rostral part). SVnc: sinus of ventral nasal concha (dorsal part). SVnc’: sinus of ventral nasal concha (ventral part). T: tongue. V: vomer. Vn: vomeronasal organ. Vnc: ventral nasal concha. Vnm: ventral nasal meatus. Zp: zygomatic process.

**Figure 10 vetsci-11-00611-f010:**
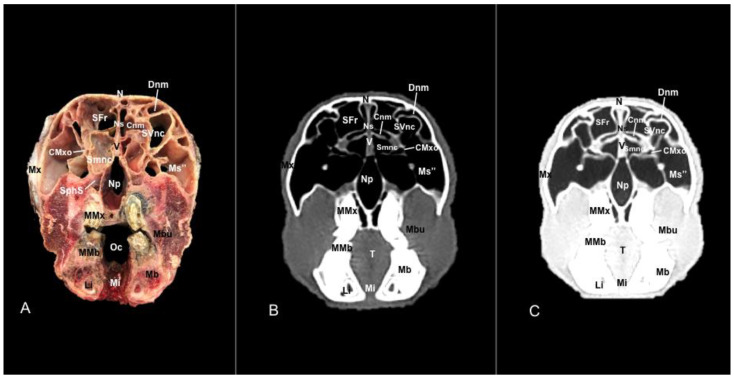
Anatomical section (**A**), bone window (**B**), and pulmonary window (**C**) CT transverse images of a crested porcupine’s nasal cavity at the level of the middle nasal concha, corresponding to line IX in Figure 1. Cnm: common nasal meatus. CMxo: conchomaxillary opening. Dnm: dorsal nasal meatus. Li: lower incisive tooth. Mi: mylohyoid muscle. Mx: maxilla. Mb: mandible. Mbu: buccinator muscle. Ms’’: caudal maxillary sinus. N: nasal bone. Np: nasopharynx. Ns: nasal septum. MMx: first molar tooth (maxillar). MMb: first molar teeth (mandibular). Oc: oral cavity. SFr: sinus of frontal bone (rostral part). Smnc: sinus of midle nasal concha. SphS: sphenopalatine sinus (palatine part). SVnc: sinus of ventral nasal concha. T: tongue. V: vomer.

**Figure 11 vetsci-11-00611-f011:**
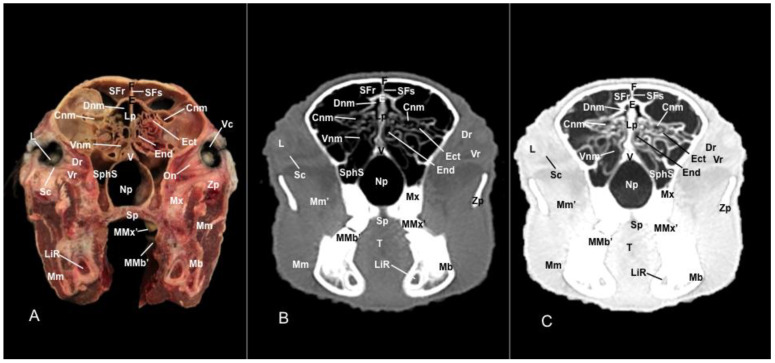
Anatomical section (**A**), bone window (**B**), and pulmonary window (**C**) CT transverse images of a crested porcupine’s nasal cavity at the level of the eyeball, corresponding to line X in Figure 1. Cnm: common nasal meatus. Dnm: dorsal nasal meatus. Dr: dorsal rectus muscle of the eye. E: ethmoid bone (tectorial plate). Ect: ectoturbinate. End: endoturbinate. F: frontal bone. L: lens. LiR: lower incisive root. Lp: perpendicular plate (ethmoid bone). Mb: mandible. Mm: masseter muscle (superficial portion). Mm’: masseter muscle (deeper portion). Mx: maxilla. MMx’: second molar tooth (maxillar). MMb’: second molar tooth (mandibular). Np: nasopharynx. Sc: sclera. SFs: septum of frontal sinuses. SFr: sinus of frontal bone (rostral part). SphS: sphenopalatine sinus (palatine part). T: tongue. V: vomer. Vc: vitreous chamber. Vnm: ventral nasal meatus. Vr: ventral rectus muscle of the eye. Zp: zygomatic process.

**Figure 12 vetsci-11-00611-f012:**
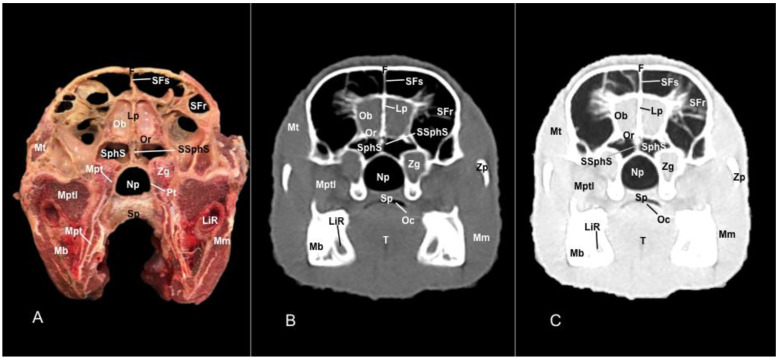
Anatomical section (**A**), bone window (**B**), and pulmonary window (**C**) CT transverse images of a crested porcupine’s nasal cavity at the level of the olfactory bulb, corresponding to line XI in Figure 1. F: frontal bone. LiR: lower incisive root. Lp: perpendicular plate (ethmoid bone). Mb: mandible. Mm: masseter muscle. Mt: temporalis muscle. Mpt: medial pterygoid muscle. Mptl: lateral pterygoid muscle. Np: nasopharynx. Ob: olfactory bulb. Or: olfactory recess. SFr: sinus of frontal bone (caudolateral part). SFs: septum of frontal sinuses. Sp: soft palate. SphS: sphenopalatine sinus (sphenoidal part). SSphS: septum of sphenopalatine sinus. T: tongue. Zg: zygomatic gland. Zp: zygomatic process.

## Data Availability

The information is available at https://accedacris.ulpgc.es/ (accessed on 24 September 2024).

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
