# Peer review of "Study of the Normal Crested Porcupine (Hystrix cristata) Nasal Cavity and Paranasal Sinuses by Cross-Sectional Anatomy and Computed Tomography"

_vetsci, 2024, doi:10.3390/vetsci11120611_

Round 1
Reviewer 1 Report
Comments and Suggestions for Authors
This paper discusses “Study of the Normal Crested Porcupine (Hystrix cristata) Nasal Cavity and Paranasal Sinuses by Cross‐Sectional Anatomy and Computed Tomography”. In general, the results of the paper confirm the rationality and correctness of the experimental method, which is consistent with the research goal. However, there are still some questions that hope the author to answer. Some comments or suggestions are as follows:
1. Why the Normal Crested Porcupine was chosen as the experimental animal
2. What are the special advantages of CT over MRI, and why not choose the two methods for comparison?
3. Is there anything special about the difference between Normal Crested Porcupine female and male samples?
4. The deficiencies and perspectives of the manuscript can be listed separately at the end of the manuscript
Author Response
Dear Reviewer,
We appreciate all your suggestions, which have helped to improve the manuscript.
Comment 1: Why the Normal Crested Porcupine was chosen as the experimental animal
Response 1: The rodent's nasal cavity is an anatomical structure with multiple functions related to the respiratory and olfactory systems. Therefore, it is responsible for heating, humidifying, and filtering the inspired air and correctly directing it in the respiratory airways. Considering the large size of this animal and its biology, eating plant material that they smell from the ground, we wanted to perform a deep description of this cavity and check any specific anatomical differences compared to mammals, including other rodents.
Comment 2: What are the special advantages of CT over MRI, and why not choose the two methods for comparison?
Response 2: Since the nasal cavities and paranasal sinuses do not contain soft-tissue structures other than the mucosa, using different CT windows is quite helpful in evaluating the normal structures that compose this cavity. CT is considered the best imaging technique for an initial evaluation of the nasal cavity due to the excellent visualization of the bony limits and their extensions. Other advantages of CT over MRI are lower cost and less time to conduct an examination. In contrast to CT, MRI allows differentiation between the nasal mucosa and other soft tissues or fluids, which can be very valuable in interpreting of pathological conditions. In our work, we were working with animals without nasal pathologies. Thus, this fact and the combination of different CT windows and anatomical cross-sections provided excellent detail of this cavity.
Comment 3: Is there anything special about the difference between Normal Crested Porcupine female and male samples?
Response 3: Despite some papers described that distribution and quality of food
resources is suggested as the factor responsible for the evolution of sexual size dimorphism in favour of females in the crested porcupine (Giorgio Pigozzi (1987) Female‐biased sexual size dimorphism in the crested porcupine (Hystrix cristata L.), Italian Journal of Zoology, 54:3, 255-259), we did not observe any essential difference between normal Crested Porcupine females and male nasal cavity samples. Nonetheless, further studies with more samples should be done to better evaluate this possible difference. This information is presented in the discussion section.
Comment 4: The deficiencies and perspectives of the manuscript can be listed separately at the end of the manuscript.
Response 4: Following your suggestion, we have listed the deficiencies, and added the possible perspectives at the end of the manuscript, before the conclusions section.
Reviewer 2 Report
Comments and Suggestions for Authors
Dear authors,
I have read your manuscript on the porcupine's nasal cavity that was illustrated by means of gross anatomical and CT sections with interest. I have listed some remarks and suggestions that could ameliorate the quality of the manuscript.
Simple summary and abstract: These should be extended. What is the rationale for the study? Where does this species live? Why is it important to study the nasal cavity? What could the results be used for?
Introduction
Line 19: species name (Hystrix cristata) should be in italics
Lines 35-37: This sentence should be rephrased. “It occurs mainly in Europe and Africa because these animals are still hunted…” So, the porcupine lives in Europe and Africa because it is hunted there?
Line44: It should be mentioned earlier that a synonym for the crested porcupine is “hystrik”.
Line 54: Remove “, as noted by Bruno 54 and Riccardi”
Line 60: premolars
Line 67: advanced imaging procedures
Line 72-77 = last paragraph of the introduction: What is the rationale for the study? It has to be more than just because the nasal cavity was not yet studied in this species. Rabbits and rodents are important pet and experimental animals. In these species, it could be valuable to spend time and money investigating their nasal cavities. Does this also apply to porcupines? Why do you compare the porcupine with the rabbit and rodents? Why do we need the information? What could the results be used for?
Materials and methods
Lines 81-82: “The animals died from causes unrelated to this region.” How did they die? Were they euthanized?
Line 90: “two CT windows” You describe three in the following lines.
Line 98-99: “transverse anatomical cross‐sections” It is not clear that you made anatomical slices from the carcasses.
Line 116: approximately
Line 119: bone window (B) and lung window (C) CT transverse images (Idem for the next legends)
General remark regarding the figure legend: anatomical terms should be in either English or Latin, not a combination. Some terms now begin with a capital letter, most do not. This should be uniform.
Line 122: teeth à tooth (Idem next legends)
Line 127: accessory
Fig. 3: Vlnc is not in the figure legend
Line 160 (and further figure legends): Choana à choanae
Line 175: what is the malar bone? Is the os malaris a specific skull bone in the porcupine?
Line 180: no capitals for m. mylohyoideus and m. buccinator. Better is to choose for uniform English terminology, thus mylohyoid muscle and buccinator muscle
Line 188: Is the m. rectus dorsalis a specific muscle in the porcupine? Idem for Vr. It is not the m. rectus capitis dorsalis?
Line 191: maxillary bone or simply maxilla instead of maxilar bone
Line 194: zygomatic process of malar bone? (Idem line 204)
3.1. Anatomical sections
I would propose to put the Latin terms in italics without capital
Line 206: “this cavity” What cavity? The nasal cavity.
Line 219: dorso- and ventrolaterally
Line 229-230: braincase could be replaced by neurocranium
Discussion
First paragraph contains content that should be included in the introduction, especially regarding the aims and rationale of the study.
Finally, the quality of the language should be increased to the standards of scientific/academic English. Here and there, double spacing is present and a full stop lacks. The font style should also be uniform.
Comments on the Quality of English Language
The English is not of high academic/scientific quality. However, it is understandable.
Author Response
Comment: Simple summary and abstract: These should be extended. What is the rationale for the study? Where does this species live? Why is it important to study the nasal cavity? What could the results be used for?
Response: As you recommend, we have extended the simple summary and the abstract, including the information requested.
Comment: Line 19: species name (Hystrix cristata) should be in italics
Response: Agree, we have changed it.
Comment: Lines 35-37: This sentence should be rephrased. “It occurs mainly in Europe and Africa because these animals are still hunted…” So, the porcupine lives in Europe and Africa because it is hunted there?
Response: We have clarified this sentence in ordert to explain the reasons of the inclusion in the IUCN.
Comment: Line 44: It should be mentioned earlier that a synonym for the crested porcupine is “hystrik”.
Response : As you recommend, we have mentioned this synonym earlier.
Comment: Line 54: Remove “, as noted by Bruno 54 and Riccardi”.
Response : As you recommend, we have removed this sentence.
Comment: Line 60: premolars.
Response: Following your suggestion, we have replaced "premolar" by "premolars".
Comment: Line 67: advanced imaging procedures.
Response: Agree, we have changed it.
Comment: Line 72-77 = last paragraph of the introduction: What is the rationale for the study? It has to be more than just because the nasal cavity was not yet studied in this species. Rabbits and rodents are important pet and experimental animals. In these species, it could be valuable to spend time and money investigating their nasal cavities. Does this also apply to porcupines? Why do you compare the porcupine with the rabbit and rodents? Why do we need the information? What could the results be used for?
Following your recommendation, we have added other reasons that could explain the rationale for the study. Therefore, we have included "The rodent's nasal cavity is an anatomical structure with multiple functions related to the respiratory and olfactory systems. It is responsible for heating, humidifying, filtering the inspired air and correctly directing it in the respiratory airways [22]. Considering the large size of this animal and its biology, eating plant material that they smell from the ground, we wanted to perform a deep description of this cavity and check any specific anatomical differences compared to mammals, including other rodents. This information could help to better understand its respiratory and olfactory functions.
Material and methods
Comment: Lines 81-82: “The animals died from causes unrelated to this region.” How did they die? Were they euthanized?
Response: the animals were euthanized for reasons beyond our study, we have added this specific information in the section.
Comment: Line 90: “two CT windows” You describe three in the following lines.
Response: Agree, we have changed it.
Comment: Line 98-99: “transverse anatomical cross‐sections” It is not clear that you made anatomical slices from the carcasses.
Response: we have clarified this concern, adding specific information.
Comment: Line 116: approximately
Response: we have redone the sentence.
Comment: Line 119: bone window (B) and lung window (C) CT transverse images (Idem for the next legends).
Response: As you recommend, we have redone the figure legends.
Comment: General remark regarding the figure legend: anatomical terms should be in either English or Latin, not a combination. Some terms now begin with a capital letter, most do not. This should be uniform.
Response: As you suggested, and concerning English or Latin, as well as with the capital letters, we have uniformed all the terms.
Comment: Line 122: teeth à tooth (Idem next legends).
Response: Agree, we have changed this legend in all figures.
Comment: Line 127: accessory.
Response: Agree, we have corrected it.
Comment: Fig. 3: Vlnc is not in the figure legend.
Response: We have included Vlnc "ventral lateral nasal cartilage" in the legends.
Comment: Line 160 (and further figure legends): Choana à choanae
Response: we have replaced "Choana" by "choanae".
Comment: Line 175: what is the malar bone? Is the os malaris a specific skull bone in the porcupine?
Response: This bone is well developed in crested porcupines and greater cane rat. However, we have relabelled this structure as zygomatic process.
Comment: Line 180: no capitals for m. mylohyoideus and m. buccinator. Better is to choose for uniform English terminology, thus mylohyoid muscle and buccinator muscle.
Response: As you recommend, we have changed to uniform english terminology.
Comment: Line 188: Is the m. rectus dorsalis a specific muscle in the porcupine? Idem for Vr. It is not the m. rectus capitis dorsalis?
Response: We apologize for this, here we are labelling the dorsal and ventral rectus muscles of the eye, which are striated muscles of the eyeball that project rostrally towards the sclera.
Comment: Line 191: maxillary bone or simply maxilla instead of maxilar bone.
Response: we have replaced by "maxilla".
Comment: Line 194: zygomatic process of malar bone? (Idem line 204)
Response: we have replaced by "zygomatic process".
3.1. Anatomical sections
Comment: Line 206: “this cavity” What cavity? The nasal cavity.
Response: Accordingly, it concerns the nasal cavity.
Comment: Line 219: dorso- and ventrolaterally.
Response: Agree, we have changed it as you recommend.
Comment: Line 229-230: braincase could be replaced by neurocranium
Response: Accordingly, we have replaced it.
Comment: Discussion
First paragraph contains content that should be included in the introduction, especially regarding the aims and rationale of the study.
Response: As you suggest, we have moved part of this content to the introduction section.
Finally, we have tried to increase the quality of the English language. Moreover, we have corrected the double spacing and uniformed the font style.
Reviewer 3 Report
Comments and Suggestions for Authors
Dear Authors, please correct it and make it clearer which are marked.

Author Response
Dear Reviewer,
We appreciate these suggestions since they have been quite helpful to improve our manuscript
Comment 1: please correct it and make it clearer which are marked.
Response 1: following your suggestions, we have included all the suggested changes along the manuscript, which are highlighted with track changes